# Mapping maternal and infant health in Morocco: A global scoping review of themes, gaps, and the "unseen" in the published health research literature, 2000–2022

Ellen Amster[1]*, Ghazal Jessani[2], Gauri Gupta[2], Oksana Hlyva[1], Charlene Rae[1]

1 Department of Family Medicine, McMaster University, Hamilton, Ontario, Canada, 2 Michael G. DeGroote School of Medicine, McMaster University, Hamilton, Ontario, Canada

* amstere@mcmaster.ca

**Data Availability Statement:** Data collected from four publicly available databases: OVID:MEDLINE, Embase, APA PsycINFO, EBESCO: CINAHL.

## Abstract

Global efforts to reduce Maternal Mortality Rates (MMR) have been significant, but researchers are exploring new approaches to address stalled progress and enduring health inequities. This scoping review offers an analytic synthesis of maternal and infant health (MIH) research in the low-middle income North African Islamic country of Morocco over 22 years, a mapping of the themes, research gaps, geographies, and methodologies, 2000–2022. Morocco is an official MIH success story with excellent health indicators, yet indicators do not address local contexts, gender issues, or health disparities. To understand how medical research has reflected social reality over the past 22 years, we explored not just what is known, but how it is known, where it is known, what remained unseen, and why. Four databases were searched: OVID: MEDLINE, Embase, APA PsycINFO, and EBSCO: CINAHL. 4590 abstracts were identified, 3131 abstracts screened, and 402 full MIH articles and 128 sub-group articles identified and subject to data extraction. The 402 full MIH articles were subject to qualitative thematic analysis, classified by 34 primary research themes and explored especially for gender, health equity, and methodology. Findings included significant geographic research disparities; four regions were the location of 75% of research and many regions remained virtually "unseen" by research. The best-equipped urban public hospitals in higher-income regions produced the most research, creating an urban, hospital-based research perspective. Maternal health articles predominated, often >50% more than articles published about infant health. Infants studied were mostly neonates. Socially marginalized women were often invisible to research, as were private healthcare, NGO care (non-governmental civic organizations), and healthcare in community. In articles, researchers recommended new policies, new laws, health system reform, and government actions to advocate for patients. Three solutions emerged to broaden the research perspective: increase geographic breadth, address missing topics and populations, and embrace interdisciplinary methods.

Relevant data for articles extracted provided with DOI in Supplement 1 and S3 Table. GIS data used for maps from publicly-available data sets, named in references with URLs.

**Funding:** The authors received no specific funding for this work.

**Competing interests:** The authors have declared that no competing interests exist.

## Introduction

Global progress to reduce maternal mortality rates (MMR) has been significant, but stalled progress [1] has shifted attention from provision of basic services to new questions—the quality of maternal and infant health care (MIH), the fragility and function of health systems, "patient-centered care," and persistent racial, regional, socioeconomic, and gender-based health inequities [2–6]. To meet these challenges, researchers have been reconceptualizing methodological frameworks to see MIH holistically [2], big-picture [3], and in interdisciplinary ways to "see the unseen" and provide comprehensive evidence for decision-makers [4]. Medical, evidence-based ways of knowing may miss important contextual factors, and the emphasis on health indicators can incentivize countries to pursue a national political "numbers game" of maternal death statistics rather than develop an applied focus on woman-centered care [7].

Morocco, a low-middle income North African Islamic country, is an exemplary official MIH success story. Morocco is one of ten low- to-middle income countries considered to have met the Millennium Development Goal (MDG) 5 threshold of consistent -5.5 annual rate of change (ARC) and the Sustainable Development Goal (SDG) 3.1 target MMR of 70 [8]. Further, Morocco promised to decrease infant mortality by 2/3 by 2015 [9]. The most recent (2018) national population health survey showed a 35% decrease in maternal mortality between 2010 and 2016 [10]. Other Moroccan national indicators are summarized in S1 Table. These statistics reflect Morocco's national plans and policies, which include improved national coverage of emergency obstetric care after 1995, upscaled training of health personnel (midwives especially) [11], implementation of a series of health system "action plans" with maternal health emphasis, reduction of barriers to care through a Free Deliveries and Caesarean Policy in public facilities [12,13], and the creation of a national health insurance program for the lowest-income citizens, the Régime d'Assistance Médicale (RAMed), implemented nationally in 2012 [13].

And yet important health disparities hide behind excellent MIH global health indicators, disparities highlighted by recent news stories from Moroccan society and healthcare professionals. Political unrest erupted in the north of Morocco in 2017 (the "Hirak Rif" movement), in part over demands for greater health infrastructure. Moroccan physicians exposed poor conditions in public hospitals on a 2014 Facebook page (now removed) [14,15], and midwives [16], physicians [17,18], and medical students [19] have organized protests and mass strike activity in 2019 to protest health system issues. There is evidence that progress on maternal mortality slowed, and a 2015 confidential internal report described most maternal deaths as occurring within the healthcare system [20,21]. Political debate erupted in the Moroccan press over public perceptions of maternal mortality numbers: "The opposition took this up and said, 'Thank God there are international agencies who will tell us the truth about our women who are dying...'"[7]. Official indicators are useful to monitor progress, but MIH research is essential to provide data in context, provide a fine-grained picture of MIH realities in Morocco, and help improve health outcomes [22].

The aim of this review is to map the field of MIH research in Morocco—what is known, how it is known, and what remains "unseen"—to build a resource for MIH researchers and suggest future directions for MIH research. To meet this objective, we undertook a global scoping review of the published MIH health literature about Morocco over a 22-year period (2000–2022). Global mapping of the literature answers recent calls to "overcome disciplinary silos" and narrow specializations in MIH, to "identify persistent and critical knowledge gaps" [23] and use a gender analytical lens [5,6,24]. To explore how health research connected with society, we focused especially on three sub-questions: **gender**, how researchers approached

women and gender, **equity**, how researchers addressed the systemic aspects of health disparities [25] and North/South equity in the research enterprise, and **methodology**, how methods connected medical data to social context. Women-centered and intersectional analyses are often not well-represented in MIH research in low- and middle-income countries [5]. Women's perspectives are needed, especially because gender inequities have important impacts on MIH health outcomes [6,26]. Interdisciplinary and mixed-methods approaches have been recommended to improve MIH research, MIH care, and MIH policy [4,27–30]. Through an overview of themes, gaps, methodologies, and geographies, this global mapping approach will allow researchers to see what questions have been asked (or not asked), what biases and blind spots exist, and where the lacunae may lie—what topics and people remain invisible to health research.

## Methods

### Scoping review

An initial search of OVID found no previous scoping review examining the current status of maternal and infant health research in Morocco. This scoping review was conducted using the Arksey and O'Malley methodological framework [31]. Four databases were searched without timeframe limits in December 2022 including: OVID: MEDLINE (1946 to present), Embase (1947 to present), APA PsycINFO (1800 to present), and EBSCO: CINAHL (1937 to present). The search strategy was developed with consultation from a Librarian at McMaster University (S2 Table). The initial search was run to capture the entire evolution of MIH in Morocco. However, upon review, authors determined that only articles published in 2000 or later reflected current Moroccan health issues and MDG policy priorities. Thus, only articles published in 2000 and onwards were included in the review for analysis; pre-2000 articles were screened for a future historical bibliography of MIH research in Morocco (and not included). Abstracts were included if the study was conducted in the Moroccan population and involved women, infants, maternal-focused health professionals (e.g., midwives), or health system (e.g., neonatal intensive care unit, health insurance); focused on an aspect of maternal (e.g., reproductive health, prenatal, pregnancy, birth, post-natal), neonatal (<30 days of age) or infant (1 month to 1 year of age) health. Exclusion criteria included language other than English, Arabic, or French, systematic reviews, unpublished work, or conference abstracts. Letters to the editors or desk reviews were eligible if qualitative or quantitative data were included or cases were described. All abstracts were uploaded to Covidence systematic review software (Veritas Health Innovation, Melbourne, Australia) and duplicates removed. Two authors, EA and CR, reviewed the titles and abstracts independently; disagreements were resolved by further discussion by screeners. Full text review was conducted in Covidence by EA and CR, and articles were included if they met the inclusion criteria described above. Articles were separated into two groups: 'full text' articles, focused solely on MIH topics or populations, and 'sub-group' articles, focused on non-MIH topics and populations but which included mothers or infants as a patient population sub-group within a larger, non-MIH study population. A hand search of citations from included full-text articles was conducted to discover relevant articles meeting inclusion criteria which were not captured in the database search.

Full text articles published in 2000 or later were extracted fully. A data extraction sheet was created in Excel (Microsoft Office 365) with variables including: general study information, study location, population, and outcome (for case study only). Studies were also classified for the level of data collection (local, regional, provincial, national, global), focus of the study, perspective of the data, and data sources. A detailed list of extraction variables is provided in S3 Table. EA and CR tested the data extraction sheet by extracting 5 articles and comparing

results to ensure completeness and consistency of data extraction. Authors in pairs continued to extract and compare data until consistency was achieved between extractors. All remaining articles were extracted individually. Sub-group articles underwent a partial extraction focusing on study location, proportion of the study population consisting of mothers or infants, health condition, and study focus (S3 Table). The sub-group extraction sheet was tested using the same method as for the full-text extraction sheet. Data extraction was done identically for French and English articles. However, as only one author (EA) is French-English bilingual, standardization of data extraction between authors for French language articles was not possible. CR reviewed extracted data and addressed any questions from extractors.

After data extraction was complete, EA applied qualitative inductive thematic analysis, (see Braun and Clarke) [32,33] to the 402 full-text articles. Preliminary themes were developed inductively from article abstracts. Articles were then organized chronologically by preliminary theme and subject to deep, full-text analysis by EA. Themes were iteratively and inductively revised and articles recategorized until thematic saturation was reached; i.e., when authors EA and CR agreed that the 34 identified themes proved coherent and meaningful. Each article was coded with a primary theme and as many secondary themes as needed to describe it; EA and CR discussed thematic classification of articles to ensure consistency and agreement.

Within each primary theme, EA organized articles chronologically, subjected them to qualitative description analysis (see Sandelowski) [34], and synthesized the principal ideas, research questions, topics, and methods of each theme's articles into one-paragraph summaries. EA and CR then met to review and resolve any outstanding data issues.

## Analysis

Data were summarized using descriptive statistics in SPSS version 26.0 (IBM Corporation, Armonk NY, USA). Geographical data (region, cities) were mapped using ArcGIS software, Version Pro 3.1.3 (ESRI, Redlands, California, USA). Five open-source Geographic Information System (GIS) mapping layers [35–39] were used to generate distribution maps representing extracted location data from full-text MIH articles (city/region) and to create a heat map of public health facilities in Morocco.

For thematic analysis of full-text articles, three critical research questions were also considered: gender, health research equity, and methodology. Gender analysis asked whether researchers centered women's voices, preferences, and life-course, what gendered assumptions were made by researchers, and how health research considered women's intersectional social, legal, and economic status. Women included patients, parents, and health professionals. For health equity in Morocco, we asked how researchers studied the rural/urban divide, geographic inequities, health disparities, and Moroccan social challenges. Outside Morocco, we considered North/South research equity—whether Moroccan researchers guided international research collaboration and how accessible research publications were (i.e. open access).

Primary, secondary, tertiary, and quaternary themes were weighted proportionally and represented in a thematic network visualization with Gephi software. This method was used to illustrate the relative importance of thematic research topics and the connections between them. Communities, (i.e. subset of nodes that have more connection within the group than with the rest of the network), were identified using the Louvain method analysis [40,41]. In the network, nodes were represented by theme labels with label size proportionate to the number of times a theme appeared as primary in the review. Edges represented links between themes within articles, with primary themes as the source and relationships weighted as follows: primary to secondary 1, primary to tertiary 0.5, and primary to quaternary 0.25. Thicker lines represent a greater connection between themes.

## Results

### Primary MIH articles

A total of 4,590 abstracts were identified from the search, with 1459 duplicates removed. EA and CR screened 3,131 abstracts, excluding 1659 irrelevant abstracts (Fig 1). During full-text screening, 203 articles were published before 2000 and thus excluded, but set aside for a future historical review. A total of 528 articles were identified for extraction, and 5 were excluded (see Fig 1). Additionally, 7 relevant articles were identified from a hand search of citations from included full-text articles. The final review included 402 fully extracted full text articles and 128 sub-group articles, the latter extracted with fewer variables. See bibliography of full-text articles organized by primary research theme in S1 Text. Relevant data for each full-text article are provided in S4 Table.

Descriptive statistics for all full MIH articles (n = 402) by two time periods (2000–2011) and (2012–2022) are displayed in Table 1. There was a threefold increase in MIH research publication from the first eleven years (n = 93) to the second eleven years (n = 309). Maternal health-focused articles greatly outnumbered infant health-focused articles, often by more than 50% in any given year (S1 Fig). Infants studied in the literature were mostly neonates, as the study population was mostly younger than 30 days old and often younger than 18 days old (S2 Fig). Amazigh patients or the Amazigh languages were not mentioned in 92.3% of studies (Table 1).

**The research enterprise.**   Most first authors were Moroccan (78.4% overall), with an increase from 66.7% in 2000–2011 to 81.9% in 2012–2022. Europeans represented 9.7% of first authors and North Americans were 7.5%. Open-access publications increased from 8.6% to 54.4% over the 22 years (43.8% of total), and languages of publication were majority English (69.1%) and French (30.9%), with English increasing over time (Table 1). No articles published in Arabic were identified (n = 0). The open-access Pan African Medical Journal was the most-published journal (12.2%) followed by Journal de Pédiatrie et de Puériculture (5.2%).

The majority of research was hospital-based (79.9%) and focused predominantly at the regional or local level (76.1%), with 14.2% at the national level and only 1.2% engaging multiple levels of the healthcare system. While 14.2% of articles were national in scope, only 9.2% of articles collected national-level data or multiregional data or used preexisting national data sets (e.g., (DHS) Demographic and Health Surveys, etc.). Multi-country international studies that included Morocco comprised 10.2% of studies. Most studies were based in hospital/health facilities, and only a minority were based outside of health structures in community (9.2%) or located in both hospital and community (5.2%) (Table 1). Of health facility-based studies, most were conducted within tertiary hospitals (48.5%). Two public tertiary hospitals in particular dominated research production; from 2000–2011, 23.7% of research was conducted at the Centre Hospitalier Universitaire (CHU) Ibn Rochd in Casablanca (13.4% overall), and in 2012–2022, 18.1% of research was conducted at the CHU Ibn Sina in Rabat (16.9% overall), (S5 Table). Only 6.5% of studies included multiple health facilities. Government research institutes (e.g. Institut National d'Hygiène) were the site of 8.0% of research, followed by primary care clinics (7.5%) and second-level provincial hospitals (4.5%). (Table 1)

Study designs were primarily cross-sectional (27.4%), patient case studies (21.6%), and retrospective studies of patient files/chart review (14.2%), followed by cohort studies (6.7%) and secondary analysis of data (6.7%). Few studies used qualitative methods (5.0%), mixed-methods (2.5%), or bench/laboratory (3.0%) methods. Methodologies applied only in the 2012–2022 period include implementation studies (1.5%), randomized controlled trials (1.0%), protocols (0.25%), ethnopharmacological studies (0.5%), descriptive (1.5%), case-control (2.5%), and modelling (0.25%). Data collection focused primarily at the patient level (66.9%), with less

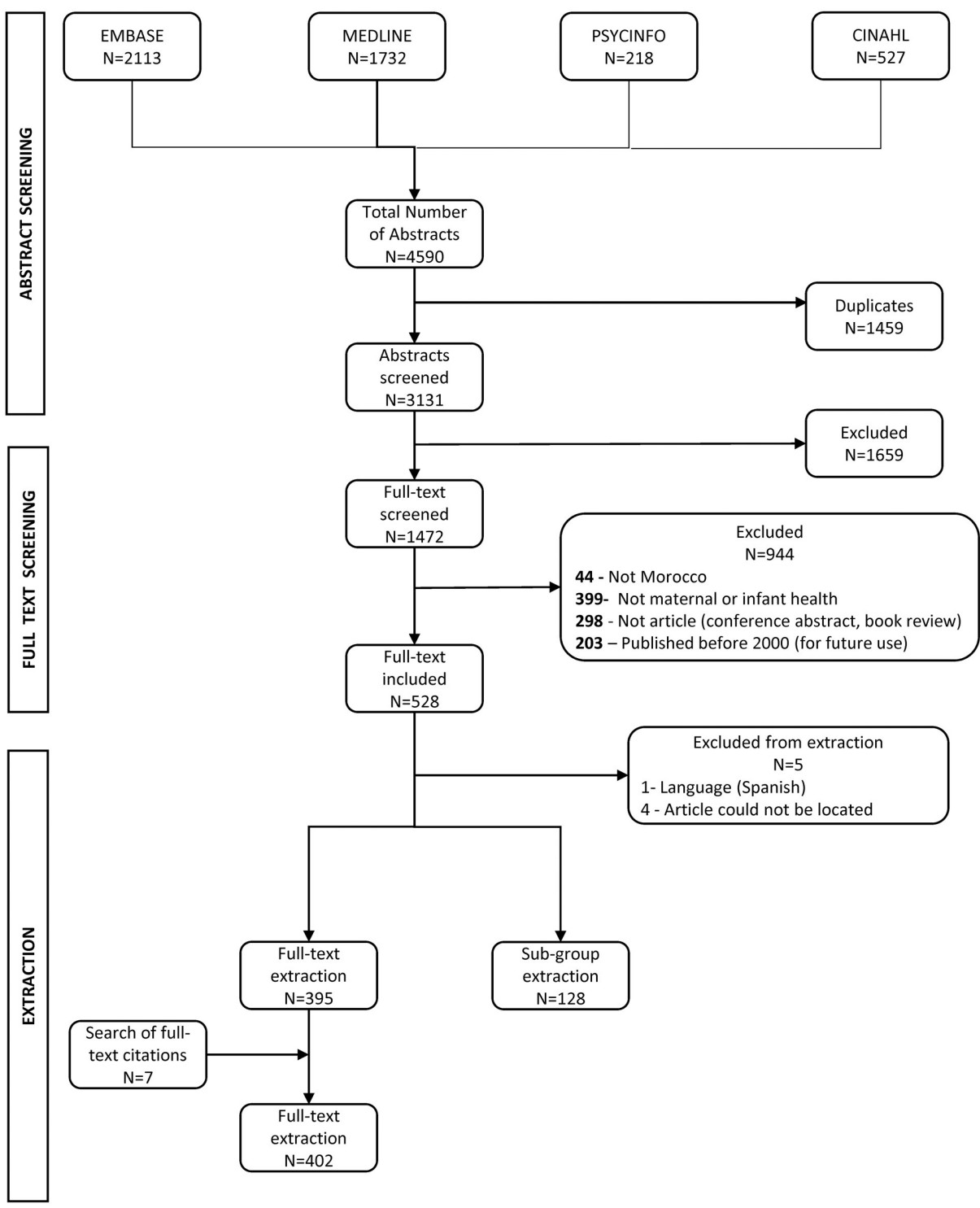

**Fig 1. PRISMA diagram.**

**Table 1. Summary statistics of full-text MIH articles.**

| | | Time period | | | | Total | |
| --- | --- | --- | --- | --- | --- | --- | --- |
| | | 2000–2011 (n = 93) | | 2012–2022 (n = 309) | | (N = 402) | |
| | | n | % | n | % | n | % |
| Language | English | 51 | 54.8 | 227 | 73.5 | 278 | 69.2 |
| | French | 42 | 45.2 | 82 | 26.5 | 124 | 30.9 |
| | Arabic | 0 | 0.0 | 0 | 0.0 | 0 | 0.0 |
| Open Access | Yes | 8 | 8.6 | 168 | 54.4 | 176 | 43.8 |
| Country affiliation of Primary Author | Morocco | 62 | 66.7 | 253 | 81.9 | 315 | 78.4 |
| | Africa | 3 | 3.2 | 3 | 1.0 | 6 | 1.5 |
| | Asia Pacific | 1 | 1.1 | 4 | 1.3 | 5 | 1.2 |
| | Europe | 12 | 12.9 | 27 | 8.7 | 39 | 9.7 |
| | Middle East | 0 | 0.0 | 1 | 0.3 | 1 | 0.3 |
| | North America | 13 | 14.0 | 17 | 5.5 | 30 | 7.5 |
| | South America | 1 | 1.1 | 2 | 0.7 | 3 | 0.8 |
| | Working group | 1 | 1.1 | 2 | 0.7 | 3 | 0.8 |
| Scope of study | International | 12 | 12.9 | 29 | 9.4 | 41 | 10.2 |
| | National | 16 | 17.2 | 31 | 10.0 | 57 | 14.2 |
| | Regional or local level | 62 | 66.6 | 244 | 79.0 | 306 | 76.1 |
| | Multiple levels | 2 | 2.2 | 3 | 1.0 | 5 | 1.2 |
| | Not applicable | 1 | 1.1 | 2 | 0.7 | 3 | 0.8 |
| Design | Bench | 3 | 3.2 | 9 | 2.9 | 12 | 3.0 |
| | Case study | 16 | 17.2 | 71 | 23.0 | 87 | 21.6 |
| | Case-Control | 0 | 0.0 | 10 | 3.2 | 10 | 2.5 |
| | Chart review | 20 | 21.5 | 37 | 12.0 | 57 | 14.2 |
| | Cohort | 5 | 5.4 | 22 | 7.1 | 27 | 6.7 |
| | Cross-sectional | 20 | 21.5 | 90 | 29.1 | 110 | 27.4 |
| | Descriptive | 0 | 0.0 | 6 | 1.9 | 6 | 1.5 |
| | Ethnopharmacological | 0 | 0.0 | 2 | 0.7 | 2 | 0.5 |
| | Implementation study | 0 | 0.0 | 6 | 1.9 | 6 | 1.5 |
| | Mixed Methods | 1 | 1.1 | 9 | 2.9 | 10 | 2.5 |
| | Modelling | 0 | 0.0 | 1 | 0.3 | 1 | 0.3 |
| | Multiple | 2 | 2.2 | 3 | 1.0 | 5 | 1.2 |
| | Observational | 1 | 1.1 | 4 | 1.3 | 5 | 1.2 |
| | Protocol | 0 | 0.0 | 1 | 0.3 | 1 | 0.3 |
| | Qualitative | 6 | 6.5 | 14 | 4.5 | 20 | 5.0 |
| | RCT | 0 | 0.0 | 4 | 1.3 | 4 | 1.0 |
| | Desk review | 4 | 4.3 | 8 | 2.6 | 12 | 3.0 |
| | Secondary analysis | 15 | 16.1 | 12 | 3.9 | 27 | 6.7 |
| Health facility-based vs community-based sample | Both | 8 | 8.6 | 13 | 4.2 | 21 | 5.2 |
| | Community | 15 | 16.1 | 22 | 7.1 | 37 | 9.2 |
| | Health facility | 64 | 68.8 | 257 | 83.2 | 321 | 79.9 |
| | Other | 6 | 6.5 | 17 | 5.5 | 23 | 5.7 |

(*Continued*)

**Table 1.** (Continued)

| | | Time period | | | | Total | |
|---|---|---|---|---|---|---|---|
| | | 2000–2011 (n = 93) | | 2012–2022 (n = 309) | | (N = 402) | |
| | | n | % | n | % | n | % |
| Region of data collection | National data collected | 15 | 16.1 | 22 | 7.1 | 37 | 9.2 |
| | Casablanca-Settat | 27 | 29.0 | 49 | 15.9 | 76 | 18.9 |
| | Drâa-Tafilalet | 1 | 1.1 | 2 | 0.6 | 3 | 0.7 |
| | Fès-Meknès | 3 | 3.2 | 32 | 10.4 | 35 | 8.7 |
| | Gharb Chrarda Bni Hssen | 0 | 0.0 | 1 | 0.3 | 1 | 0.2 |
| | Guelmim-Oued Noun | 0 | 0.0 | 1 | 0.3 | 1 | 0.2 |
| | L'Oriental | 1 | 1.1 | 11 | 3.6 | 12 | 3.0 |
| | Marrakech-Safi | 13 | 14.0 | 57 | 18.4 | 70 | 17.4 |
| | Rabat-Salé-Kénitra | 22 | 23.7 | 91 | 29.4 | 113 | 28.1 |
| | Souss-Massa | 0 | 0.0 | 2 | 0.6 | 2 | 0.5 |
| | Tanger-Tétouan-Al Hoceïma | 0 | 0.0 | 5 | 1.6 | 5 | 1.2 |
| | Western Sahara | 0 | 0.0 | 1 | 0.3 | 1 | 0.2 |
| | Multiple regions reported | 5 | 5.4 | 15 | 4.9 | 20 | 5.0 |
| | Location(s) not reported | 3 | 3.2 | 12 | 3.9 | 15 | 3.7 |
| | Not applicable | 3 | 3.2 | 8 | 2.6 | 11 | 2.7 |
| Type of health facility where study conducted | Tertiary hospital | 40 | 43.0 | 155 | 50.2 | 195 | 48.5 |
| | Government Research Institute (e.g. Institut National d'Hygiène) | 9 | 9.7 | 23 | 7.4 | 32 | 8.0 |
| | Provincial hospital | 3 | 3.2 | 15 | 4.9 | 18 | 4.5 |
| | Regional hospital | 3 | 3.2 | 6 | 1.9 | 9 | 2.2 |
| | Clinic (including mobile) | 5 | 5.4 | 25 | 8.1 | 30 | 7.5 |
| | Multiple | 6 | 6.5 | 34 | 11.0 | 40 | 10.0 |
| | Other | 0 | 0.0 | 1 | 0.3 | 1 | 0.3 |
| | Not applicable | 23 | 24.7 | 39 | 12.6 | 62 | 15.4 |
| | Not reported | 4 | 4.3 | 11 | 3.6 | 15 | 3.7 |
| Amazigh Population | No | 83 | 89.2 | 288 | 93.2 | 371 | 92.3 |
| | Yes | 6 | 6.5 | 14 | 4.5 | 20 | 5.0 |
| | Not reported—but study in Amazigh population area | 4 | 4.3 | 4 | 1.3 | 11 | 2.7 |
| Focus of the data collected (level) | Laboratory data | 1 | 1.1 | 2 | 0.7 | 3 | 0.8 |
| | Patient-level data | 59 | 63.4 | 210 | 68.0 | 269 | 66.9 |
| | Population-level data | 13 | 14.0 | 29 | 9.4 | 42 | 10.5 |
| | Provider-level data | 4 | 4.3 | 16 | 5.2 | 20 | 5.0 |
| | System-level data | 16 | 17.2 | 46 | 14.9 | 62 | 15.4 |
| | Multiple levels | 0 | 0.0 | 4 | 1.3 | 4 | 1.0 |
| | Other | 0 | 0.0 | 2 | 0.7 | 2 | 0.5 |

(*Continued*)

**Table 1.** (Continued)

| | | Time period | | | | Total | |
|---|---|---|---|---|---|---|---|
| | | 2000–2011 (n = 93) | | 2012–2022 (n = 309) | | (N = 402) | |
| | | n | % | n | % | n | % |
| Primary data source | Administrative (e.g. chart review/patient files) | 31 | 33.3 | 81 | 26.2 | 112 | 27.9 |
| | Biological sample/ Biometric measures | 12 | 12.9 | 61 | 19.7 | 73 | 18.2 |
| | Case description | 12 | 12.9 | 58 | 18.8 | 70 | 17.4 |
| | Large population survey | 12 | 12.9 | 11 | 3.6 | 23 | 5.7 |
| | Literature/desk review | 5 | 5.4 | 5 | 1.6 | 10 | 2.5 |
| | Patient/parent interview | 3 | 3.2 | 12 | 3.9 | 15 | 3.7 |
| | Patient/parents questionnaire | 12 | 13.0 | 45 | 14.6 | 57 | 14.2 |
| | Provider interview | 3 | 3.2 | 12 | 3.9 | 15 | 3.7 |
| | Provider questionnaire | 2 | 2.2 | 10 | 3.2 | 12 | 3.0 |
| | Researcher observation | 1 | 1.1 | 4 | 1.3 | 5 | 1.2 |
| | System interviews | 0 | 0.0 | 1 | 0.3 | 1 | 0.3 |
| | User email | 0 | 0.0 | 1 | 0.3 | 1 | 0.3 |
| | Workshop proceedings | 0 | 0.0 | 2 | 0.7 | 2 | 0.5 |
| | Multiple | 0 | 0.0 | 4 | 1.3 | 4 | 1.0 |
| | Other | 0 | 0.0 | 2 | 0.7 | 2 | 0.5 |

focus on population-level data (10.5%), healthcare system-level data (15.4%) or healthcare providers (5.0%) (Table 1).

**Geographic distribution of health research.** Regional distribution of research publication is displayed in Table 1, Figs 2–4 and S3 Fig. Research was primarily concentrated in four regions: Rabat-Salé-Kénitra (27.9%), Casablanca-Settat (18.9%), Marrakech-Safi (16.7%), and Fès-Meknès (9.2%). Inclusion of multiple regions occurred in 32 studies (8.0%). Article count differences between maps and figures/tables are due to the administrative disappearance of one region (Gharb Chrarda Beni Hssen) in 2015 and some multiple-region studies. Several regions were virtually un-researched over the past 22 years: Gharb Chrarda-Beni-Hssen was studied in only one publication (0.25%). Sous-Massa was in 2 publications (0.5%) and included in two multi-region studies. Drâa-Tafilalet was in 3 publications (0.75%) and included in three multi-region studies. Tangier-Tétouan-Al Hoceïma was studied in 4 articles (1.0%) and included in 10 multi-region studies. Beni-Mellal-Khénifra, a mostly Amazigh region, had 0 primary articles, though one city (Khénifra) was included in 2 multi-region studies. Other regions include l'Oriental (3.0%) and the disputed territory of Western Sahara/Laayoune-Sakia el Hamra (0.25%). The Fig 3 map shows locations of study sites in Morocco (where specified). Rabat (n = 119) predominated, followed by Casablanca (n = 72) and Marrakesh (n = 63). University hospitals (CHU) seemed to increase regional research; after the cities of Fez (2009) and Oujda (2014) received a CHU, the number of studies focused on their respective regions (Fes-Meknes, Oriental) increased. S4 Fig is a heat map showing the spatial distribution of public healthcare facilities throughout Morocco, with concentrations in the North Atlantic Rabat-Casablanca corridor.

**Thematic analysis.** Using inductive thematic analysis, we identified 34 primary research themes: Abortion, AIDS/STI, Bacterial Infection, Birth, Breastfeeding, Cancer, Diabetes, Environment, Family Planning, Genetics, Gynecology, Infant Morbidity, Infant Mortality, Infant Near-Miss, Legal, Maternal Morbidity, Maternal Mortality, Maternal Near-Miss, Midwifery, Newborn/Neonatal Health, Nutrition, Other Non-Respiratory Viruses, Parasitic Disease,

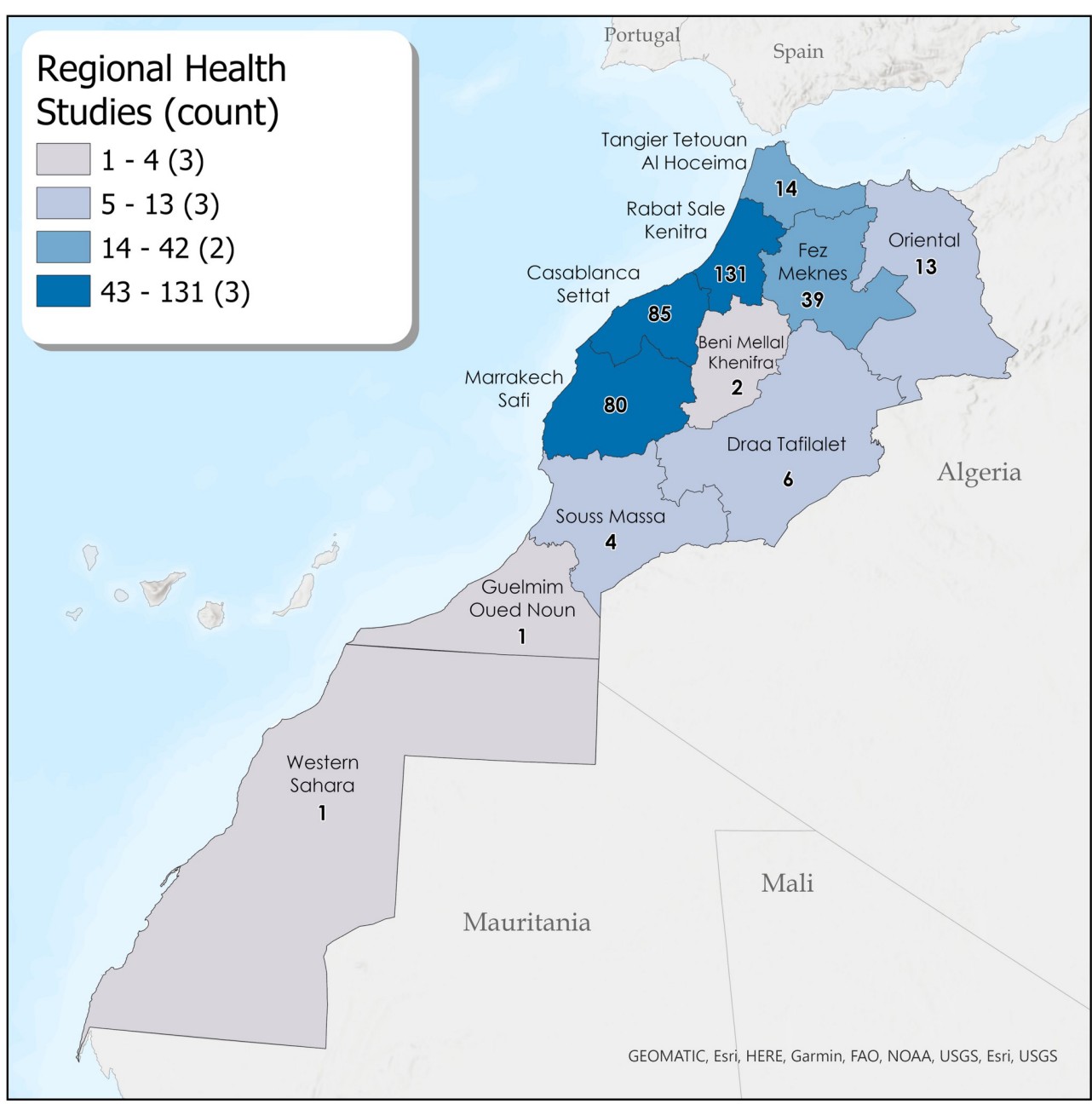

**Fig 2. Regional distribution map of study locations.** Esri, GEOMATIC, HERE, Garmin, FAO, NOAA, USGS, OpenStreetMap contributors. Maps were created using ArcGIS® software by Esri. ArcGIS® and ArcMap™ are the intellectual property of Esri and are used herein under license. Copyright Esri. All rights reserved. For more information about Esri® software, please visit www.esri.com. United Nations Office for the Coordination of Humanitarian Affairs (OCHA)–Humanitarian Data Exchange, https://data.humdata.org/dataset/cod-ab-mar; https://data.humdata.org/dataset/cod-ab-esh, CC BY 4.0.

Preeclampsia/Eclampsia, Pregnancy, Public Information/Literacy, Respiratory Virus, Rural/Amazigh, Screening for Newborns, Social Determinants of Health, Technology, The Healthcare System, Traditional Medicine, and Vaccination. A complete, thematically-organized bibliography of extracted full-text articles is provided in S1 Text.

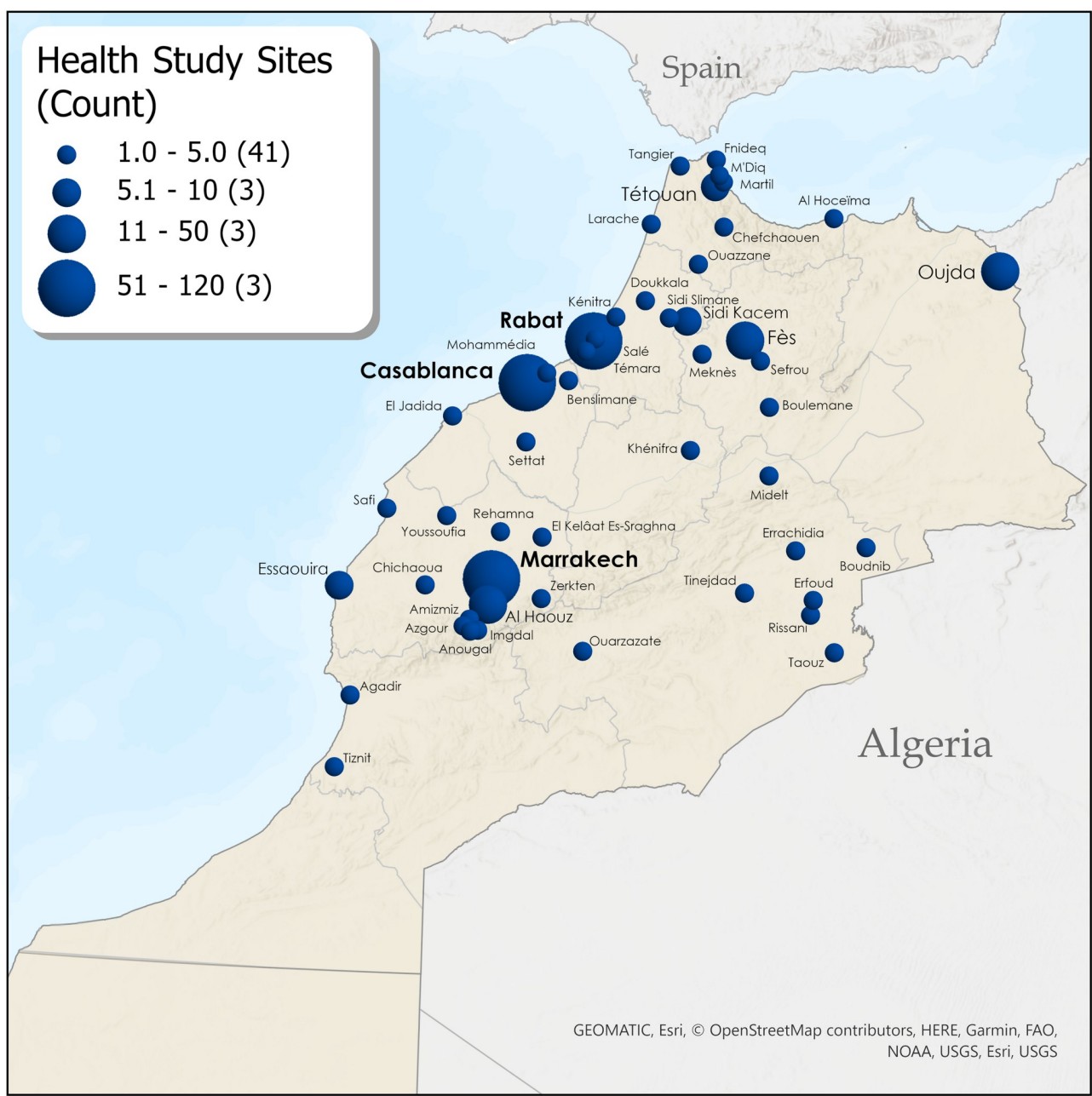

**Fig 3. Geographic distribution map of study site locations with proportional representation by number of articles.** Esri, GEOMATIC, HERE, Garmin, FAO, NOAA, USGS, OpenStreetMap contributors. Maps were created using ArcGIS® software by Esri. ArcGIS® and ArcMap™ are the intellectual property of Esri and are used herein under license. Copyright Esri. All rights reserved. For more information about Esri® software, please visit www.esri.com. United Nations Office for the Coordination of Humanitarian Affairs (OCHA)–Humanitarian Data Exchange, https://data.humdata.org/dataset/cod-ab-mar; https://data.humdata.org/dataset/cod-ab-esh; https://data.humdata.org/dataset/hotosm_mar_populated_places, CC BY 4.0.

For each theme, we used qualitative description to summarize the principal sub-topics discussed by researchers, identify changes over time, and describe the research methodologies utilized in Table 2. The percentage of the literature dedicated to each theme is shown in Fig 5 and the absolute numbers of articles per theme in S5 and S6 Figs and S6 Table show themes by

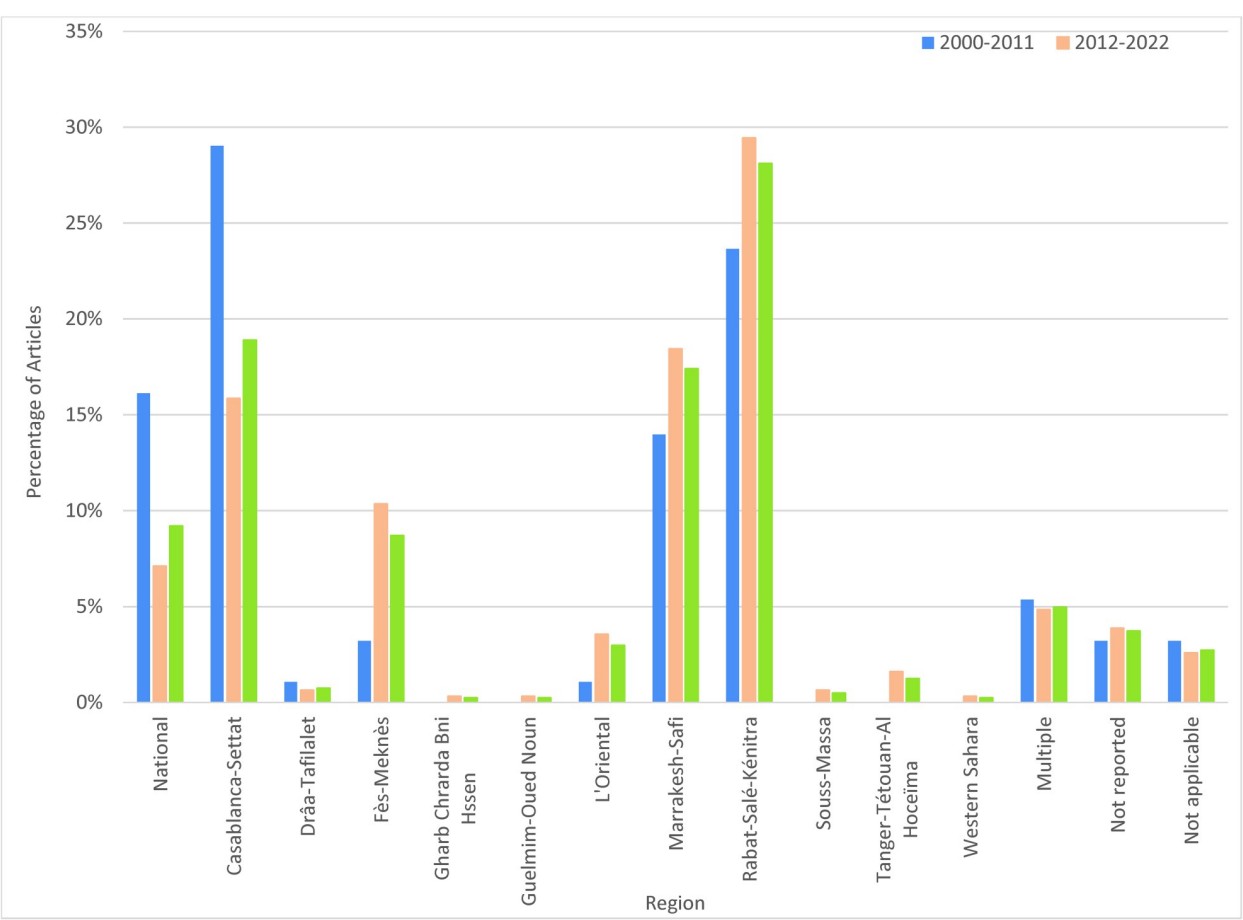

**Fig 4. Regional distribution of study locations by percentage of articles and time period.**

population focus. The first major research areas in the field (2000–2011) included Bacterial Infection, Birth, Family Planning, Genetics, Maternal Morbidity, Pregnancy, and The Healthcare System. After 2012, Family Planning declined and other themes including Maternal Mortality, Newborn/Neonatal Health, Infant Morbidity, Traditional Medicine, Breastfeeding, and Parasitic Disease gained prevalence. In 22 years, several themes remained virtually un-studied; there was only 1 AIDS study and 0 STI studies, 1 Infant Near-Miss study, 1 Legal (law-related) study, and 2 Public Education/Health Literacy (S5 Fig). Several themes (16/34) had no qualitative work. Rural/Amazigh was a patient-centered and methodologically diverse theme, but with few articles (n = 5).

To represent the relative importance and interrelations of themes in the literature, we used Gephi software to create a thematic network visualization (Fig 6).

Network analysis detected 5 "communities" (represented by label colour), representing a high number of connections between research themes. Label size reflects the prevalence of a theme. The strongest single connection in the network was between Bacterial Infection and Newborn/Neonatal Health, represented by the thickest and darkest line. Unexpected connections like Abortion-Traditional Medicine reflect context-specific issues in Morocco like the use of the traditional remedy "*harmel*" (*Peganum harmala*) as an abortifacient.

**Table 2. Summaries of the research theme areas and associated study designs.**

| Primary Theme | Summary of findings | N | Bench | Case study/series | Case-Control | Chart review | Cohort | Cross-sectional | Descriptive | Implementation study | Mixed Methods | Observational | Qualitative | RCT | Review (e.g desk) | Secondary analysis | Other | Multiple | Reference (S1 Text) |
|---|---|---|---|---|---|---|---|---|---|---|---|---|---|---|---|---|---|---|---|
| **Abortion** | Abortion engaged four topics: legal abortion restriction (defining the "Islamic" view of abortion), experiences of abortion-seeking women, emergency cases of abortion/suicide, and how criminalization of abortion produces poor health outcomes. Medical, legal, religious, and social themes were raised, and most articles contained policy recommendations. Abortion was a crime under the Moroccan Penal Code and excluded from Moroccan Sexual and Reproductive Health policy. Abortion prevalence was unknown thus statistical estimates varied quite widely. | 10 | | 3 | | 1 | | | 1 | | | | 3 | | 1 | 1 | | | 1–10 |
| **AIDS/STI** | One hospital-based review of nine cases of HIV+ women. | 1 | | | | 1 | | | | | | | | | | | | | 11 |
| **Bacterial Infection** | Bacterial infection focused on neonates and the NICU: antibiotic-resistant bacteria, nosocomial infections, iatrogenic health practices, device-associated infection (catheters and ventilators), tests to avoid antibiotic over-use, and calls for national-level antibiotic stewardship. Bacteria studied included Group B streptococcus, Enterobacteriaceae, and Acinetobacter. Other topics included maternal-infant infections, the WHO-certified elimination of neonatal tetanus in 2002, neonatal sepsis, and meningitis. The declining effectiveness of specific classes of antibiotics can be traced over the period 2000–2022. | 31 | 1 | 4 | | 8 | 6 | 10 | | | | | | | | 1 | 0 | 1 | 12–42 |

*(Continued)*

**Table 2.** (Continued)

| Primary Theme | Summary of findings | N | Bench | Case study/series | Case-Control | Chart review | Cohort | Cross-sectional | Descriptive | Implementation study | Mixed Methods | Observational | Qualitative | RCT | Review (e.g desk) | Secondary analysis | Other | Multiple | Reference (S1 Text) |
|---|---|---|---|---|---|---|---|---|---|---|---|---|---|---|---|---|---|---|---|
| **Birth** | Birth engaged emergency obstetric care availability and quality, and caesarean sections—c-section rates, fetal outcomes, maternal complications, and over-medicalization of birth. Other topics included Moroccan maternal health policies, women's uptake of facility deliveries, preventing preterm births, and improving birth outcomes. Three women-centered ethnographies explored women's beliefs around birth, care-seeking behaviours, perception of risk, decision-making, rural birth experiences, and views about traditional birth attendants. Quantitative authors called for qualitative work to examine health equity. Adolescent mothers and chorioamnionitis were also mentioned. | 19 | | 1 | 1 | 6 | | 2 | 1 | | 1 | | 2 | | | 5 | 0 | | 43–61 |
| **Breastfeeding** | Breastfeeding focused on the decline of exclusive breastfeeding in Morocco, but also women and work. Research focus shifted over time from the "unwilling" or insufficiently educated mother to mothers' working conditions, access to lactation support, family support, public information, and new mothers' perceptions, beliefs, and preferences. Other topics included breastfeeding of premature infants, infant nutrition in nutrient-deprived areas, iodine deficiency, and iodine supplementation. Of note is a randomized controlled trial of iodine supplementation in the [Amazigh] High Atlas; 10% of infants were stunted at baseline. Female healthcare professionals as breastfeeding mothers were discussed. | 17 | | 1 | | | 4 | 9 | | 1 | 1 | | | 1 | | | 0 | | 62–78 |

*(Continued)*

**Table 2.** (Continued)

| Primary Theme | Summary of findings | N | Bench | Case study/ series | Case-Control | Chart review | Cohort | Cross-sectional | Descriptive | Implementation study | Mixed Methods | Observational | Qualitative | RCT | Review (e.g desk) | Secondary analysis | Other | Multiple | Reference (S1 Text) |
|---|---|---|---|---|---|---|---|---|---|---|---|---|---|---|---|---|---|---|---|
| Cancer | Cancer and pregnancy were the focus: how pregnancy can complicate cancer, how cancer treatment can adversely affect women's fertility and fetal development, and the bioethics of medical decision-making. Oncofertility, advocacy for cancer survivors to access Ovarian Tissue Cryopreservation, and gestational trophoblastic diseases were also discussed. | 6 | 1 | | | 1 | 2 | 2 | | | | | | | | | 0 | | 79–84 |
| Diabetes | Diabetes focused on screening for gestational diabetes mellitus: how screening was applied, the need for rapid, inexpensive point-of-care tests, and healthcare professionals' knowledge, training, workload, and compliance with policy. Improved screening revealed high gestational diabetes prevalence in Morocco and mandatory national screening was recommended. Four studies include patient perspective: patient knowledge, barriers to laboratory testing, social stigma of diabetes, and difficulty of diet change at home. Researchers suggested relocating diabetes screening and uncomplicated disease management to primary care. | 8 | | | | | | 3 | | 1 | 1 | | 1 | 2 | | | 0 | | 85–92 |
| Environment | Environment focused on the presence in maternal breastmilk of toxins like lead, aluminum, cadmium, aflatoxin M1, mercury, ochratoxin, and pesticides, but also toxins in infant hair, groundwater, soil, and traditional healing plants (henna, *mkhinza*, kohl). Moroccan food legislation and the insufficient regulation of food industries were indirectly implicated by the presence of ochratoxin and aflatoxin in breastmilk. Also discussed were asthma prevalence and the poisonous effect of insecticide. | 9 | | 1 | 1 | | | 7 | | | | | | | | | 0 | | 93–101 |

*(Continued)*

**Table 2.** (Continued)

| Primary Theme | Summary of findings | N | Bench | Case study/ series | Case-Control | Chart review | Cohort | Cross-sectional | Descriptive | Implementation study | Mixed Methods | Observational | Qualitative | RCT | Review (e.g desk) | Secondary analysis | Other | Multiple | Reference (S1 Text) |
|---|---|---|---|---|---|---|---|---|---|---|---|---|---|---|---|---|---|---|---|
| **Family Planning** | Family planning included use of contraceptives by women, prevalence of use, and women's preferred methods. Contraception providers and pharmaceutical companies were studied. Other topics included national family planning policy, fertility patterns, shrinking Moroccan family size, infertility, and assisted reproductive technologies. After 2008, researchers focused on contraceptive access for rural, unmarried, and low-income women. NGOs, private providers, and pharmaceutical companies were found to increase access for these marginalized groups. | 13 | | | | 2 | | | | | | | 2 | | | 9 | 0 | | 102–114 |
| **Genetics** | Genetics included genetic disorders, ABO and RH factor issues, and population genetics. The authors focused on alpha and beta thalassemia, phenylketonuria (PKU), Trisomy 21, Trisomy 18, Omenn syndrome, determining the prevalence, diagnosis, and treatment of these conditions, and consanguinity in Morocco. Researchers recommended national-level mandatory newborn screening for disorders, the creation of a national bioethics committee, and government funding for genetic medicine. Challenges included the availability of specialized drugs, testing for rare diseases, and insufficient state support for disabled children. Genetic testing methods (IgD, FISH), gene mutations, biotinidase deficiency, and Beckwith-Wiedemann Syndrome were also discussed. | 23 | | 10 | | 1 | 4 | 8 | | | | | | | | | 0 | | 115–137 |
| **Gynecology** | Gynecology included a retrospective of vulvular dermatological conditions, a case of ovarian failure, and acquired arterio-venous malformation after curettage. | 3 | | 2 | | 1 | | | | | | | | | | | | | 138–140 |

*(Continued)*

**Table 2.** (Continued)

| Primary Theme | Summary of findings | N | Bench | Case study/ series | Case-Control | Chart review | Cohort | Cross-sectional | Descriptive | Implementation study | Mixed Methods | Observational | Qualitative | RCT | Review (e.g desk) | Secondary analysis | Other | Multiple | Reference (S1 Text) |
|---|---|---|---|---|---|---|---|---|---|---|---|---|---|---|---|---|---|---|---|
| **Infant Morbidity** | Infant morbidity was dominated by congenital malformations and their prevalence, especially defects of neural tube closure. Researchers identified the mother's ingestion of fenugreek and maternal folic acid deficiency as key causes, though maternal nutrition was primary. Authors recommended women's education, creation of a teratovigilance registry, prenatal screening, and genetic counseling. Infant diarrhea, colic, and allergy were discussed but from a provider perspective: what providers knew, what providers prescribed, and estimated prevalence according to providers. Other topics included urinary tract infections, chronic granulomatous disease, psoriasis, cholestasis, antibiotic overuse, and case studies. | 20 | | 7 | | 4 | 2 | 5 | | | | 1 | 1 | | | | 0 | | 141–160 |
| **Infant Mortality** | Infant mortality examined primarily perinatal death. Earlier authors attributed perinatal infant mortality to prematurity, maternal consumption of fenugreek, and congenital abnormality. Later researchers moved to a social determinants of health research approach that found correlations of infant death with poverty, rural provenance, referral from another health structure, poor delivery conditions, and neonatal respiratory distress. "Medico-legal" responsibility for infant death was also discussed. | 4 | | 1 | 1 | 1 | | 1 | | | | | | | | | 0 | | 161–164 |
| **Infant Near-Miss** | This neonatal near-miss study compared statistics from Morocco, Benin, and Burkina Faso but concluded that qualitative research is needed to explain contextual reasons for near-miss. | 1 | | | | | | 1 | | | | | | | | | 0 | | 165 |

*(Continued)*

**Table 2.** (Continued)

| Primary Theme | Summary of findings | N | Bench | Case study/series | Case-Control | Chart review | Cohort | Cross-sectional | Descriptive | Implementation study | Mixed Methods | Observational | Qualitative | RCT | Review (e.g desk) | Secondary analysis | Other | Multiple | Reference (S1 Text) |
|---|---|---|---|---|---|---|---|---|---|---|---|---|---|---|---|---|---|---|---|
| **Legal** | Legal explored the interface of public hospitals with police, law, state, and medical case documentation in a qualitative study of single mothers. The single mother has ambiguous legal status because sex outside marriage was criminalized. Documentation generated by the hospital can harm or hide these vulnerable patients. | 1 | | | | | | | | | | 1 | | | | | 0 | | 166 |
| **Maternal Morbidity** | Maternal morbidity covered topics from kidney injury to depression, dental health, hemorrhage, life course after morbidity, and STIs. However, definitions of maternal morbidity varied across studies. One approach used hospital-based diagnosis immediately after birth and concluded that women's illiteracy and high parity were key variables. Other researchers studied a period up to 8+ months after birth, compared rural/urban and different regions, considered mental and physical effects of near-miss on women, used interviews, and evaluated doctor-patient communication. Longer-term study designs cited much higher maternal morbidity prevalence, poor communication between doctors and patients, life course consequences for women (mental, physical, economic, social) of maternal morbidity, geographic and structural health disparities, and hidden morbidities (STIs). | 24 | | 7 | | 7 | 3 | 6 | | | 1 | | | | | | 0 | | 167–190 |

*(Continued)*

**Table 2.** (Continued)

| Primary Theme | Summary of findings | N | Bench | Case study/series | Case-Control | Chart review | Cohort | Cross-sectional | Descriptive | Implementation study | Mixed Methods | Observational | Qualitative | RCT | Review (e.g desk) | Secondary analysis | Other | Multiple | Reference (S1 Text) |
|---|---|---|---|---|---|---|---|---|---|---|---|---|---|---|---|---|---|---|---|
| **Maternal Mortality** | Maternal mortality publications began in 2013 about the national maternal death surveillance system (MDSS)—its development, implementation, results, and limitations. By 2013, most reported maternal deaths were found to be avoidable and to occur within the public healthcare system. However, maternal deaths were underreported to the MDSS. Reasons included absence of data on home births, onerous reporting procedures, providers' fear of liability, exclusion of "late maternal death," rural populations' reluctance to report to Ministry of Interior, and financial sustainability. Also discussed were Moroccan national policies, rural/urban disparities, staff views of MDSS, and verbal autopsy. Several authors offered policy recommendations. | 9 | | 1 | | 1 | | | 2 | 1 | 1 | 1 | | | | 2 | 0 | | 191–199 |
| **Maternal Near-Miss** | Maternal near-miss began with practitioners implementing audits in Morocco, Benin, Cote d'Ivoire, Mali, and Ghana; Morocco scaled up the audit process. Of the 5 countries, only Morocco had audit meetings without patient feedback and without midwives in leadership roles. Incidence, characteristics, and determinants of near-miss were studied, with a focus on quality of care, demographics of near-miss mothers, and avoidable factors. Researchers contrasted women's perceptions of near-miss with health professionals' views. Policy recommendations, women-centered care, and rural women's challenges were discussed. Researchers emphasized the need for patient interviews to understand maternal near-miss. | 5 | | | 1 | 2 | | | | | | | 1 | | | | 0 | 1 | 200–204 |

*(Continued)*

**Table 2.** (Continued)

| Primary Theme | Summary of findings | N | Bench | Case study/ series | Case-Control | Chart review | Cohort | Cross-sectional | Descriptive | Implementation study | Mixed Methods | Observational | Qualitative | RCT | Review (e.g desk) | Secondary analysis | Other | Multiple | Reference (S1 Text) |
|---|---|---|---|---|---|---|---|---|---|---|---|---|---|---|---|---|---|---|---|
| **Midwifery** | Midwifery focused on the midwife—gender and social context of her work, difficult working conditions (especially in rural areas), her empowerment, education, training, and mentoring. Researchers evaluated the Moroccan Ministry of Health's implementation of an action plan to scale up midwifery. Gender analyses addressed patriarchy in the healthcare system and in society's views of midwives. Qualitative work examined experiences of midwives and of birthing patients. Critiques were offered of "technical" medical care and how midwives were integrated to healthcare teams. Other topics included rural/urban disparities, a new Western Saharan refugee midwifery school, emergency obstetric care, and traditional birth attendants. | 10 | | | | | | | | 2 | 1 | | 3 | | 3 | | 0 | 1 | 205–214 |
| **Newborn/ Neonatal Health** | Newborn/Neonatal articles included neonatal intensive care in Morocco, neonatal morbidity, and congenital malformations (see also Infant Morbidity). For neonatal intensive care, authors discussed Moroccan hospital capacity, resource allocation, therapies, nosocomial infection, technology, and device-associated complications. Neonatal morbidity focused on low birth weight—its prevalence, risk factors, and relationship to maternal nutrition and morbidity. Secondary topics included perinatal asphyxia, neonatal anemia, and macrosomia. Also mentioned were icterus, hemorrhagic syndrome, cysts, pleural effusion, collodion syndrome, adrenal hematoma, thrombosis, and cholestasis. Birth conditions, the challenges of data collection, surgical procedures, maternal morbidities, and rural environments were also discussed. Case studies predominated. | 28 | | 10 | 5 | 5 | 4 | 4 | | | | | | | | | 0 | | 215–242 |

*(Continued)*

**Table 2.** (Continued)

| Primary Theme | Summary of findings | N | Bench | Case study/series | Case-Control | Chart review | Cohort | Cross-sectional | Descriptive | Implementation study | Mixed Methods | Observational | Qualitative | RCT | Review (e.g desk) | Secondary analysis | Other | Multiple | Reference (S1 Text) |
|---|---|---|---|---|---|---|---|---|---|---|---|---|---|---|---|---|---|---|---|
| **Nutrition** | Nutrition included population studies, mother-infant pairs, and community-based studies, examining geographic disparities (rural/urban) and social determinants. Articles explored dietary intake, anemia in pregnant women, obesity, nutritional deficiencies in iodine, iron, folate, vitamin D, and vitamin A in mothers and infants, methods of nutritional supplementation, and government policies to enrich food staples. The Moroccan diet and traditional practices were discussed. Researchers considered how maternal nutritional deficits contribute to maternal morbidity (anemia, gestational diabetes, hydatiform mole, hemorrhage, preeclampsia), and infant morbidity (congenital conditions, low birth weight, stunting, hypothyroidism). Authors advocated region-specific nutrition policies and better state regulation of food industries. | 16 | 1 | | | 1 | | 11 | | | | | | 1 | | 1 | 0 | 1 | 243–258 |
| **Other Non-Respiratory Viruses** | Non-respiratory virus articles examined a nosocomial epidemic of rotavirus in a NICU and a case study of bullous varicella in an infant. | 2 | | 1 | | 1 | | | | | | | | | | | 0 | | 259–260 |
| **Parasitic Disease** | Three endemic parasitic diseases were discussed: hydatid cysts caused by Echinococcus granulosus in women, visceral and cutaneous leishmaniasis in infants, and toxoplasmosis during pregnancy. Toxoplasmosis predominated—prevalence, risk factors, igG avidity, impact on the fetus, women's knowledge, and health professionals' knowledge. Agriculture, hygiene, animal and environmental vectors, health education for mothers, cost and availability of testing, and government policies were discussed. Morocco was a pilot trial site for a point-of-care fingerstick toxoplasmosis screening technology. | 16 | | 4 | | 2 | | 8 | | | | | | | | 2 | | 0 | | 261–276 |

*(Continued)*

**Table 2.** (Continued)

| Primary Theme | Summary of findings | N | Bench | Case study/series | Case-Control | Chart review | Cohort | Cross-sectional | Descriptive | Implementation study | Mixed Methods | Observational | Qualitative | RCT | Review (e.g desk) | Secondary analysis | Other | Multiple | Reference (S1 Text) |
|---|---|---|---|---|---|---|---|---|---|---|---|---|---|---|---|---|---|---|---|
| Preeclampsia/ Eclampsia | The prevalence, risk factors, clinical characteristics, evolution, prognosis, emergency management, underdiagnosis, and clinical evolution of these hypertensive disorders were considered. These hospital-based articles focused on preeclampsia, eclampsia, HELLP syndrome, and complications—hemorrhage, renal failure, perinatal morbidity, perinatal death, liver hematoma, posterior reversible encephalopathy syndrome, and hypertensive retinopathy. Most preeclampsia/eclampsia patients received no prenatal care, and some cases never reached the hospital. Training of primary care nurses, midwives, and community workers in "miniPIERS" (Preeclampsia Integrated Estimate of Risk) tool for timely diagnosis was recommended by one researcher. | 10 | | 4 | 1 | 4 | 1 | | | | | | | | | | 0 | | 277–286 |
| Pregnancy | Pregnancy morbidities were a primary theme: pregnancy and placenta accreta, PROM (premature rupture of membranes), Von Willebrand factor, uterine rupture, renal complications, mental health, physical violence, miscarriage, uterine scarring after c-section, and case studies. Neonatal morbidity and mortality and the ethical challenges of decision-making for physicians were also explored. A second theme was pregnant women's health behaviour: uptake of prenatal care, use of psycho-active substances, self-medication, use of sunscreen, and sexual activity. Abnormal pregnancies were a third theme—ectopic, heterotopic, and hydatiform mole—management, risk factors, prevalence, and evolution. Single mothers, gender-based violence, and marital relationships were also mentioned. Case studies predominated. | 37 | 1 | 22 | | 5 | 1 | 8 | | | | | | | | | 0 | | 287–323 |

*(Continued)*

**Table 2.** (Continued)

| Primary Theme | Summary of findings | N | Bench | Case study/ series | Case-Control | Chart review | Cohort | Cross-sectional | Descriptive | Implementation study | Mixed Methods | Observational | Qualitative | RCT | Review (e.g desk) | Secondary analysis | Other | Multiple | Reference (S1 Text) |
|---|---|---|---|---|---|---|---|---|---|---|---|---|---|---|---|---|---|---|---|
| **Public Information/ Literacy** | Public health education considered what patients and families know about the nature, risk and progress of an illness (infant bronchiolitis or hypertensive disorders of pregnancy, HDP). The HDP was the only article in the entire review to discuss pedagogy and use women's feedback for a specific health education strategy. Almost all research assumed the parent to be educated is the mother. | 2 | | | | | | 1 | | | | | 1 | | | | 0 | | 324–325 |
| **Respiratory Virus** | Respiratory virus was dominated by COVID-19—clinical and epidemiological characteristics of infected pregnant and postpartum women. A Marrakesh CHU public maternity ward reorganized to meet COVID-19 protocols, which revealed decreased availability of obstetric tertiary care, resource scarcity, and acute stress on staff. Other topics included H1N1 flu infection in pregnant and postpartum women, PCR tests to distinguish viral from bacterial pneumonia, and pertussis. | 7 | | 1 | | 1 | | 2 | 1 | | 1 | 1 | | | | | 0 | | 326–332 |

(Continued)

**Table 2.** (Continued)

| Primary Theme | Summary of findings | N | Bench | Case study/series | Case-Control | Chart review | Cohort | Cross-sectional | Descriptive | Implementation study | Mixed Methods | Observational | Qualitative | RCT | Review (e.g desk) | Secondary analysis | Other | Multiple | Reference (S1 Text) |
|---|---|---|---|---|---|---|---|---|---|---|---|---|---|---|---|---|---|---|---|
| **Rural/ Amazigh** | Rural articles were methodologically innovative and included qualitative, community-based, "action-research," determinants of health, woman-centered, and appreciative inquiry approaches, as well as hospital data. Themes included maternal morbidities, antenatal care, health promotion, and local uptake of maternal care. Research centered women patients: their experiences, evaluation of care quality (especially primary care), health knowledge, barriers to care, and suggestions for improvement. Men's attitudes about [women's] reproductive health and traditional birth attendants were also discussed. Some interviews were conducted in Amazigh (Berber) language (Tashilhit and Tamazight) with women, but also with men and local associations (civil society NGOs). Interviewers were often medical students. | 5 | | | | | | 2 | | 1 | 1 | | 1 | | | | 0 | | 333–337 |
| **Screening for Newborns** | Screening for newborns articles focused on the challenges, importance, ethics, system requirements, and implementation barriers to introducing newborn and prenatal screening in Morocco, which did not exist in 2022. Congenital hypothyroidism was studied in two pilot screening studies. Screening pilots were also conducted for Downs syndrome, Becker muscular dystrophy, and congenital heart disease. Acceptability of screening to parents, medical decision-making, screening technologies, and pan-African collaboration were explored. Abortion law, nutrition, and health policy were also discussed. Researchers advocated for national newborn screening, a national registry of congenital anomalies, and a National Bioethics Committee. | 7 | | 1 | | | | 3 | 1 | | 1 | | | | | 1 | 0 | | 338–344 |

*(Continued)*

**Table 2.** (Continued)

| Primary Theme | Summary of findings | N | Bench | Case study/ series | Case-Control | Chart review | Cohort | Cross-sectional | Descriptive | Implementation study | Mixed Methods | Observational | Qualitative | RCT | Review (e.g desk) | Secondary analysis | Other | Multiple | Reference (S1 Text) |
|---|---|---|---|---|---|---|---|---|---|---|---|---|---|---|---|---|---|---|---|
| **Social Determinants of Health** | Here social determinants focused on population health and how gender, social norms, economics, geographic disparities, poverty, law, government policies, family, water quality, food security, infrastructure, population growth, women's legal status, women's education, and women's financial independence impact health outcomes for mothers and infants. Outcomes of MIH primary care-level health interventions, a national shift in development policy from economy to society (INDH), and neoliberalism in motherhood were explored. Researchers highlighted urban/rural, rich/poor, and regional health disparities and critiqued simple approaches to "women's empowerment." Women's fertility choices, the cost of living, family code (Moudawanna) reforms, health insurance, Islam, political instability, domestic violence, and NGOs were mentioned. Authors offered policy recommendations. | 5 | | | | | | | | | | | 1 | | | 4 | 0 | | 345–349 |
| **Technology** | Technology articles concerned the feasibility, expense, development, and implementation of new health technologies in Morocco. These included biological sample analysis, health information systems, data privacy, digital printing, enhanced communications, and mobile phone apps. Specific topics included a logistics management information system supplying contraceptives, ultrasound, point-of-care whole blood testing, gene sequencing, patient apps, mobile personal health records, and digitally printed trays for cleft palate. Researchers examined technology's potential to improve clinical care, reduce barriers, and collect data, but also to add expense, burden healthcare workers, and allow multinational corporations inappropriate access to personal data. | 7 | | 1 | | | | 3 | | | | | | | 2 | | 1 | | 350–356 |

*(Continued)*

**Table 2.** (Continued)

| Primary Theme | Summary of findings | N | Bench | Case study/ series | Case-Control | Chart review | Cohort | Cross-sectional | Descriptive | Implementation study | Mixed Methods | Observational | Qualitative | RCT | Review (e.g desk) | Secondary analysis | Other | Multiple | Reference (S1 Text) |
|---|---|---|---|---|---|---|---|---|---|---|---|---|---|---|---|---|---|---|---|
| **The Healthcare system** | Moroccan healthcare system articles ranged in topic: Moroccan MIH health policies and national action plans, health system strengths and strategic priorities, models of proposed policies, health insurance, and the experiences of healthcare providers, administrators, and hospitals. Also discussed were health regulatory legislation, pharmaceuticals, contextual barriers to policy implementation, emergency obstetrical care, and quality of care. Fee exemption policy for delivery and caesarean section was studied through mixed-methods—not only impact on MMR, IMR, and patient uptake, but also quality of care, hospital funding, staff workload, and hidden expenses for patients and families. Health economics was absent save one activity-based cost analysis of a Tiznit hospital. The Moroccan system was compared to healthcare in other low and middle-income countries. Public vs. private healthcare was discussed, but the private sector provided no data and was largely unregulated. Women's uptake of prenatal and postnatal care, family health expenses, and a need for woman-centered care were discussed. Researchers also considered health education for the public, patient-provider relationships, NGOs, medical abortion, primary care, and infertility. Many researchers offered policy recommendations. | 20 | | | | | | 8 | | | 1 | 1 | 2 | | 4 | 2 | 1 | 1 | 357–376 |

*(Continued)*

**Table 2.** (Continued)

| Primary Theme | Summary of findings | N | Bench | Case study/ series | Case-Control | Chart review | Cohort | Cross-sectional | Descriptive | Implementation study | Mixed Methods | Observational | Qualitative | RCT | Review (e.g desk) | Secondary analysis | Other | Multiple | Reference (S1 Text) |
|---|---|---|---|---|---|---|---|---|---|---|---|---|---|---|---|---|---|---|---|
| **Traditional medicine** | Traditional medicine focused on plant and mineral pharmacopeia used by Moroccan families and traditional healers. Research approaches were toxicological, pharmaceutical, ethnopharmacological, and ethnographic. The toxic, teratogenic, and abortifacient aspects of materia medica were considered through case histories, bench studies, animal trials, and the Moroccan Pharmacovigilance Center. Several plants were tested for pharmacodynamic properties, especially against multidrug resistant bacteria. Remedies used for gynecological, obstetrical, and infant health remedies were given in detail. The transmission of traditional medical knowledge from generation to generation was explored. Accidental poisoning, legal regulation, environmental pollutants, older female traditional healers (*ferraga*), and traditional cosmetics were discussed. | 20 | 8 | 5 | | 1 | | 3 | | | | | 1 | | | | 2 | | 377–396 |
| **Vaccination** | Vaccination examined especially rubella, its seroprevalence among women of childbearing age, risk factors, and the Moroccan national vaccination strategy. Also discussed were Hepatitis B, vaccination delay of premature infants, and low uptake of H1N1 and rubella vaccines among pregnant women. Cultural perceptions of vaccines by the Moroccan public and barriers to vaccination in rural environments were also discussed. | 6 | | | | 1 | | 3 | | | | | 1 | | | 1 | 0 | | 397–402 |

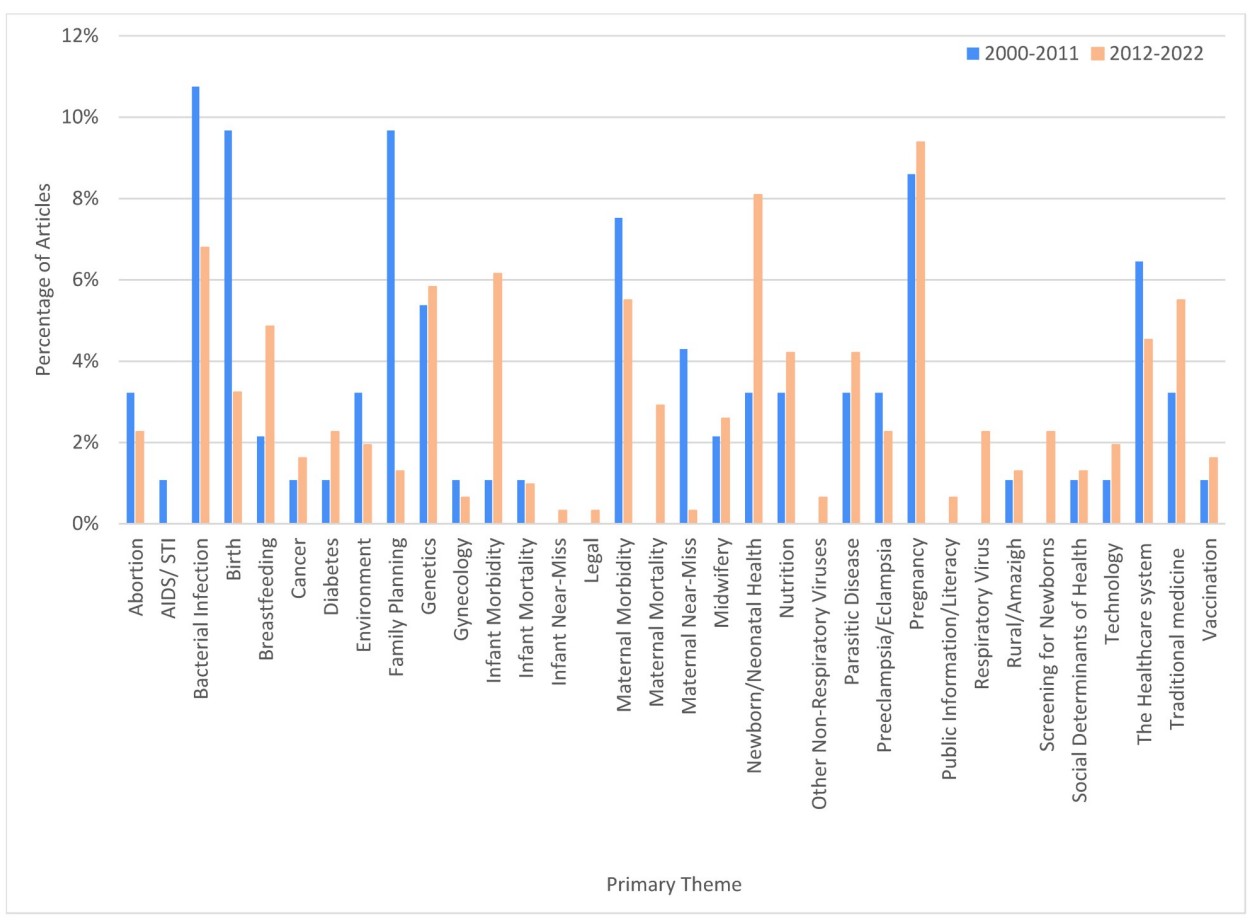

**Fig 5. Distribution of articles by primary theme and time period.**

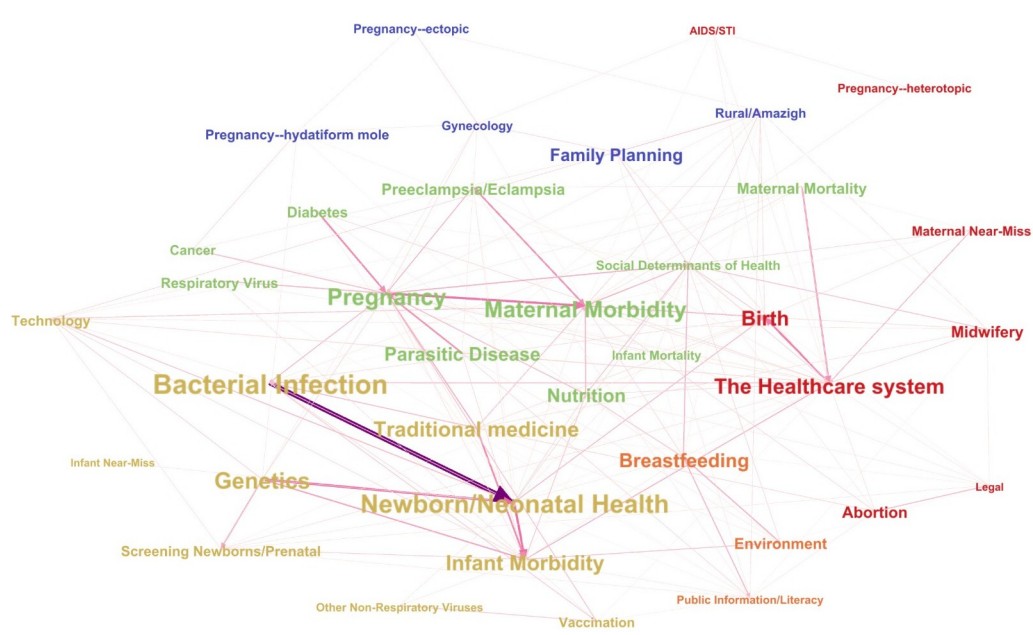

**Fig 6. Thematic network visualization of Moroccan MIH research themes, 2000–2022.**

### MIH sub-group population articles

We also analyzed a second group of articles (n = 128) separately, which included MIH populations as sub-groups within a larger non-MIH study population. Sub-group articles are described by publication date, research theme, and percentage of the study population that were mothers, infants, or both, in S7 and S8 Figs and S7 Table. Infants were the primary MIH sub-population represented within these other studies, which focused mostly on bacterial infection, genetics, respiratory virus, and the healthcare system.

## Discussion

This global scoping review mapped the published MIH research literature 2000–2022 and found substantial geographies, topics, and populations that were "unseen" by MIH research. We consider how future health research might broaden to encompass missing Moroccan populations, locations, and social realities, thus better integrating health research with Moroccan society. The discussion below provides an overview of what is known in MIH research, but also how it is known, where it is known (geographically), how researchers have addressed gender, health equity, and methodology, some existing research gaps, and a synthesis of policy and research recommendations from the authors of included research articles.

### Summary of overall findings—What is known

MIH published research about Morocco grew significantly over 22 years, with a threefold increase from the first eleven years to the second. Maternal health dominated the field, reflecting the prioritization of maternal health by the Moroccan government in National Health Action Plans (2008–2012) (2012–2016), and a National Healthcare Plan (2025) to meet MDGs 4 and 5 and reduce both MMR and IMR (Infant Mortality Rates) [13,42]. Research interests also shifted with Moroccan health progress and WHO priorities, from high interest in family planning and birth in 2000–2011 to other topics in 2012–2022, potentially reflecting higher contraception uptake, decreased MMR, and a shift in Moroccan births from 1/3 under medical supervision in 2000 to 2/3 under medical supervision by 2011[43]. Birth in medical facilities did not decrease maternal mortality as much as expected, however [20]. Policymakers were surprised to learn that reported maternal deaths were mostly avoidable and found in 2009 [20] and 2015 to occur mainly within the healthcare system [21], representing a "third delay" [44] implicating the quality of care.

Women patients' experiences in care were often not captured, especially women from rural areas. Maternal health research was often hospital-based and pathology-focused rather than women-centered; even topics central to women's experience, like Pregnancy, Preeclampsia/ Eclampsia, Gynecology, and Cancer included no qualitative work or interviews. Abortion prevalence estimates remained speculative, as abortion remained criminalized under Article 449 of the Moroccan Penal Code and excluded from Moroccan Sexual and Reproductive Health Policy.

Infant health research increased between 2000–2022 but was mostly neonatal; infants studied were mostly neonates in hospital and younger than 20 days old. The Neonatal Intensive Care Unit (NICU) was a principal focus and heavily associated with bacterial infection. After 2012, researchers advocated strongly for the creation of national newborn screening programs, using economic, human rights, and population health arguments. Infant health for ages 1–12 months was little studied, especially among rural infants. Infant research is clearly needed; a 2020 randomized controlled trial in the High Atlas Mountains found 10% of infants were stunted at baseline and without improvement over nine months [45].

The Moroccan Healthcare System itself was a major research theme, with important critical analysis [46] of national MIH policies, policy implementation, health system challenges, data shortcomings, technology, and provider perspectives. Self-critique began early with a maternal near-miss study (2001) and the creation of a Maternal Death Surveillance System (MDSS) (2009) [47], reflecting the Moroccan government's willingness to face public criticism. Midwives were the only healthcare professionals specifically researched, though physicians' knowledge, attitudes, and practice were discussed. A few studies considered how maternity healthcare teams have navigated the Free Delivery and Caesarean policy and the COVID-19 pandemic [48,49].

## Research perspective and blind spots—How it is known

The "known" of Moroccan MIH originated mostly from the best-equipped urban public hospitals in higher-income regions, producing a public hospital-based, urban, and technical perspective in MIH research. Study designs were 80% hospital-based and predominantly located in Rabat, Casablanca, and Marrakesh, with two hospitals prominent—CHU Ibn Rochd in Casablanca and CHU Ibn Sina in Rabat. University hospitals (CHU) also seemed to be important for greater regional research; after a region received a CHU, studies of that region appeared to increase.

**Geographic research disparities—Where it is known.**  Research disparities by geography were significant. The Moroccan regions most disadvantaged according to regional equity studies [50] were also the least-researched from 2000–2022, constituting an effective "blind spot" in MIH research. Nearly 75% of all research focused on the four regions with the highest levels of development and income: Rabat-Salé-Kénitra, Casablanca-Settat, Marrakech-Safi, and Fès-Meknès. Several regions were virtually un-researched over 22 years, areas with large rural populations and the fewest health resources and physicians per inhabitant (S4 Fig) [51–53]: Gharb Chrarda-Beni-Hssen (absorbed as a region in 2015), Drâa-Tafilalet, Sous-Massa, and Beni-Mellal-Khénifra. Tangier-Tétouan-Al Hoceïma, population over 3.5 million, represented only 1% of studies. The disputed Western Sahara/Laayoune-Sakia el Hamra, a region of acute MIH need [54], had only one study. In the recent past, political revolt and populist electoral politics have erupted as a result of these Moroccan regional disparities [50,55]. Future research in "unseen" regions will help connect health disparities to social and political contexts and suggest how health policies might be tailored to contextual regional needs [46].

## Gender, health equity, and methodology

How researchers addressed gender, health equity, and methodology illuminates in part how medical researchers connected health issues with Moroccan social reality.

**Women and gender.**  Of the gender issues examined, researchers considered how gender inequalities influenced health behaviors, outcomes, and women's life-course [56], defining and studying "women's empowerment," violence against pregnant women, healthcare professionals' preconceptions, maternal morbidity over the life course, gendered social stigmas, gender-based work issues, and women's experiences in care. Midwifery researchers linked Moroccan women's status to health—the 2004 Family Code, women's representation in parliament, the Ministry of the Family, and media representation [57,58]. Patient voice was a focus in recent Maternal Morbidity work (Assarag et al) [59]. Women's economic dependence and workforce participation emerged as key MIH research issues [60,61] meriting future exploration.

Unspoken gender assumptions could sometimes lead to reductive research conclusions, for example, rural women's health outcomes attributed to "women's illiteracy" [62], "ignorance," or "carelessness" [63]. By contrast, structural, qualitative, and network analyses typically

included patient interviews and/or social determinants—rural women had different outcomes not from "carelessness" but more delay in care, poorer nutrition, hard physical labour, family pressure, and hidden expenses [64]. Here women's literacy was evaluated as a complex variable, with impact from birthweight [65] to empowerment [61] to care quality, "If you are not educated, they speak to you in another tone" [66]. Breastfeeding, Diabetes, and Maternal Morbidity researchers shifted focus over time from individual to structural inquiry, from the mother's personal characteristics to her work, family, and social determinants. Women-centered research is needed for effective intervention design [67–69]. Among women-centered approaches, studies considered women's mental health, household finances, geography, workload, patient experiences and values, the family context, use of traditional medicine, and complex decision-making as patients and parents. Innovative research designs included a feminist identification of women physician-researchers with patients as breastfeeding mothers [70,71], community-based "action research," [72] and "appreciative inquiry" [64].

**Health equity.** Of equity issues, researchers paid the most attention to income, correlating health outcomes to wealth quintiles. Innovative work approached income as a complex variable, such as the "hidden" costs of "free" healthcare [73]. Social Determinants articles disaggregated global Moroccan health indicators to show regional, gender-based, and income-based health disparities [51,52]. Researchers in Nutrition, Maternal Morbidity, Midwifery, Breastfeeding, and Rural/Amazigh often designed research to explore disparities and social determinants [74]. Amazigh populations and languages were not mentioned in 92% of articles, though Morocco is estimated to be 40%-50% Amazigh [75–77] and Amazigh women can be disadvantaged in healthcare [78–80].

We also considered Global North/Global South health research equity, and whether Moroccan researchers were setting the Moroccan MIH agenda. Moroccans were the majority of first authors (78.4%). In keeping with global trends, Moroccan MIH research became more accessible; research was 8.6% open access in 2000–2011 and 54% in 2012–2022. Most Moroccan MIH articles were published in English, which is not a Moroccan national language. International collaboration was impactful when extending Moroccan research priorities. For example, Moroccan genetics researchers could use European laboratory capacity [81], Moroccan teams implemented an international toxoplasmosis screening pilot [82,83], and Moroccan researchers invited foreign experts for capacity-building [84,85].

**Methodology issues.** Quantitative data and methods heavily predominated. Researchers sometimes mentioned this issue and requested future qualitative work to identify causality among variables, address human interactions in healthcare, and understand local contexts, behavior, and patient decision-making. Qualitative studies (5% of total) addressed difficult to access topics like abortion, female traditional medical practitioners, medical pluralism, and rural women's views [79,86–88]. Mixed-methods studies used qualitative work to formulate research frameworks, show viewpoints of patients, providers, and administrators, and describe hidden social dynamics. Evidence-based interventions can fail due to local social, economic, cultural, political, and gender-based factors [89]. A "close, egalitarian alliance" with social science [90] can complement quantitative work [4,27,29] to "reveal the unseen" and help to design successful, context-sensitive MMR policies in North Africa.

## Research gaps—The missing and the "unseen"

Several groups of vulnerable women were virtually invisible to health research—women who had abortions, asylum-seeking mothers, HIV+ mothers, child/adolescent maids, and single mothers. These vulnerable MIH patients experience higher levels of morbidity, mortality, violence, and barriers to care [91–93]. Legal and social marginalization affect research visibility:

abortion, sex outside of wedlock, and extra-legal migration are criminalized; child maids are sent unofficially to work in wealthy homes and can be trafficked [94]; single motherhood and HIV+ status carry heavy social stigma. These patients can also be "unseen" if they seek care from NGOs (non-governmental organizations) rather than hospitals [86,95], or if Moroccan clinicians hide their status in hospital documentation to protect them from police proceedings [96]. Rural Amazigh MIH populations also merit greater study [45,78–80,97].

Also missing were the private healthcare sector and NGOs, which both provide important maternal and infant healthcare in Morocco—MIH data came from the public sector only. Private medicine appeared indirectly; a 2013 breastfeeding study mentioned in passing that general practitioners in Marrakesh were 40% public doctors and 60% private doctors [98]. Expensive laboratory tests, bloodwork, and ultrasounds sometimes lacking in public care were described as easily available in private care [63,99,100]. Private MIH healthcare is growing rapidly in parallel to the public system, raising questions of equity, resources, and regulation [13,101,102]. NGOs were also not studied directly, though NGOs provide care to rural, low-income, and other vulnerable populations (HIV+ women, single mothers, asylum-seeking women, disabled children) in Morocco. Private and NGO care have human resource implications for the public system, as public specialists open private practice [103] and public doctors can work shifts in NGO clinics [13,95]. Traditional birth attendants (*qablat*) still provide non-biomedical MIH care and thus merit further study [64].

Future research about MIH physicians, residents, nurses, medical students, and medical schools will illuminate their professional challenges and inform MIH human resource allocation. The laboratory's role in MIH care merits exploration, including issues of capacity, territorial distribution, and patient access. MIH public health literacy, health economic analysis, and medical legislation were often recommended by researchers as solutions to health challenges, but these MIH topics were little researched; Ouasmani et al (2018) offer a model for health literacy pedagogy [104].

### Application—Researcher recommendations for change

Researchers advocated strongly for patients, new policies, new laws, changes to the Moroccan healthcare system, and government action. This synthesis of their recommendations (Table 3) offers concrete action items to policymakers and represents context-based, expert researcher opinion repeated across several domains of Moroccan MIH.

### Strengths and limitations

Strengths of the review include its broad scope, which revealed themes, researcher recommendations, gaps in the literature, and previously unseen connections between research domains. The extended time frame showed change over time, allowing policies to be followed from implementation to evaluation. Limitations of the review lay with our inclusion criteria, which included only published health research. We did not include conference proceedings, unpublished work, systematic/literature reviews, or articles not listed in health databases, thus excluding grey literature and some social science on Moroccan MIH topics. Only one data extractor was bilingual in English and French (EA), thus data extraction of French-language articles was done by EA only and could not be standardized.

### Conclusion

Three research solutions emerged to broaden the MIH research perspective in Morocco and guide future research: increase geographic breadth, address missing topics and populations, and embrace interdisciplinary methods. Researchers must receive support to broaden

**Table 3. Researcher recommendations for policies, laws, healthcare system, and government.**

| Area | Recommendation |
|---|---|
| **Patients** | Researchers advocated for greater legal and reproductive rights for women to improve MIH, more women's education, greater economic opportunities for women, workplace support and childcare, and especially MIH rural populations—healthcare, nutrition, and poverty alleviation. Centering the woman patient's point of view in research was recommended. |
| **New Policies** | Policy recommendations included creating mandatory national newborn screening for genetic and congenital disorders (congenital hypothyroidism, phenylketonuria (PKU), and thalassemia), prenatal screening for Trisomy 18 and 21, national-level diabetes and toxoplasmosis screening for pregnant women, and state support for disabled children. Alleviation of regional disparities and health policies tailored to region-specific needs were often recommended. |
| **Legal Reforms** | New legislation was a frequent request. Legal access to abortion was proposed to prevent unsafe clandestine abortions and allow decision-making in prenatal genetic screening. Violence against women legislation was presented as a "perinatal health issue" [105]. Midwifery researchers asked for new legislation defining midwives' scope of practice and limiting provider legal liability. New regulatory legislation was requested to limit toxins in the environment, give ethical guidelines for genetics research and reproductive technologies, and to protect patient data privacy in smartphone health and pregnancy apps. |
| **Health System Reform** | Researchers asked for national-level stewardship of antibiotics to prevent their overuse and to monitor drug-resistant bacteria, the creation of a National Bioethics Center, and a National Register of Teratovigilance. Also recommended were measures to strengthen primary care, incorporate new health information technologies, improve governance, equity, and communication within the healthcare system, and improve staff retention in rural environments. |
| **Government Action** | Researchers requested better state enforcement of existing legislation regulating food industries (especially salt and dried foods) to remove contaminants, alleviate nutritional deficiencies, and reduce birth defects. Several researchers recommended health education campaigns for the Moroccan public to increase MIH health literacy among women and men. |

geographic and territorial scope, to encompass neglected regions, primary and secondary care, small cities and rural areas, and health outside the hospital. To address health equity, political unrest, and provider burnout, missing themes and populations must be examined. Interdisciplinary methods are recommended to tackle research lacunae, help frame research questions, consider gender issues, see beyond the hospital, and design women-centered, context-sensitive, and community-connected interventions and policies. This overview and its thematically organized bibliography (S1 Text) will provide a foundation for future work. Focused sub-reviews on specific themes in Moroccan MIH are to follow.

## Supporting information

**S1 Fig. Counts of articles by year and population focus (maternal, infant or both).**
(DOCX)

**S2 Fig. Distribution of infant age (days) reported in articles.** *Method of reporting age varied by study; therefore, age value was approximated using reported means, medians, and midpoints of categorical age ranges.
(DOCX)

**S3 Fig. Regional distribution of study locations by article count and time period.**
(DOCX)

**S4 Fig. Heat map showing distribution of major health facilities in Morocco by region.**
(DOCX)

**S5 Fig. Count of articles by theme.**
(DOCX)

**S6 Fig. Count of articles by population focus (maternal, infant, or both).**
(DOCX)

**S7 Fig. Distribution of subgroup MIH articles by primary theme and time period.**
(DOCX)

**S8 Fig. Approximate percentage of study population composed of mothers or infants in sub-group MIH articles.**
(DOCX)

**S1 Table. Indicators of Moroccan maternal and infant health across both study time periods (2000–2010 and 2011–2022).**
(DOCX)

**S2 Table. Search strategy.**
(DOCX)

**S3 Table. Extraction variables.**
(DOCX)

**S4 Table. Relevant data for full-text articles.**
(DOCX)

**S5 Table. Hospitals, Institutions, and Other Locations where research was conducted by time period.**
(DOCX)

**S6 Table. Frequency of articles by primary theme, population focus, and time period.**
(DOCX)

**S7 Table. Summary statistics of sub-group MIH articles.**
(DOCX)

**S1 Text. Bibliography of full-text MIH articles organized by primary theme.**
(DOCX)

## Acknowledgments

We would like to thank Jeffrey Stone for geographic data analysis and cartography.

## Author Contributions

**Conceptualization:** Ellen Amster.

**Data curation:** Ellen Amster, Charlene Rae.

**Formal analysis:** Ellen Amster, Charlene Rae.

**Investigation:** Ellen Amster, Ghazal Jessani, Gauri Gupta, Oksana Hlyva, Charlene Rae.

**Methodology:** Ellen Amster, Charlene Rae.

**Project administration:** Charlene Rae.

**Resources:** Oksana Hlyva.

**Supervision:** Ellen Amster, Charlene Rae.

**Writing – original draft:** Ellen Amster, Charlene Rae.

**Writing – review & editing:** Ellen Amster, Ghazal Jessani, Gauri Gupta, Oksana Hlyva, Charlene Rae.

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
