## [Decision Letter · Decision Letter 0]

22 Jan 2024

PGPH-D-23-02479

Mapping Maternal and Infant Health in Morocco: A Global Scoping Review of Themes, Gaps, and the "Unseen" in the Published Health Research Literature, 2000-2022

Dear Dr. Amster,

Thank you for submitting your manuscript to PLOS Global Public Health. After careful consideration, we feel that it has merit but does not fully meet PLOS Global Public Health’s publication criteria as it currently stands. Therefore, we invite you to submit a revised version of the manuscript that addresses the points raised during the review process.

We look forward to receiving your revised manuscript.

Kind regards,

Ramachandran Thiruvengadam, M.D.,

Academic Editor

Journal Requirements:

Additional Editor Comments (if provided):

Reviewers' comments:

Reviewer's Responses to Questions

**Comments to the Author**

1. Does this manuscript meet PLOS Global Public Health’s publication criteria? Is the manuscript technically sound, and do the data support the conclusions? The manuscript must describe methodologically and ethically rigorous research with conclusions that are appropriately drawn based on the data presented.

Reviewer #1: Yes

Reviewer #2: Yes

2. Has the statistical analysis been performed appropriately and rigorously?

Reviewer #1: Yes

Reviewer #2: No

3. Have the authors made all data underlying the findings in their manuscript fully available (please refer to the Data Availability Statement at the start of the manuscript PDF file)?

Reviewer #1: Yes

Reviewer #2: Yes

4. Is the manuscript presented in an intelligible fashion and written in standard English?

Reviewer #1: Yes

Reviewer #2: Yes

5. Review Comments to the Author

Reviewer #1: The authors have presented a highly relevant research topic with good methods. However, some of the points have to be addressed to improve the manuscript.

Major comments

• In introduction, the authors emphasise that gender equities are not represented in MIH. How can this be better explained when maternal health is all about adolescent and pregnant women?

• If the intention was to include only post-2000 articles, why was the search run from inception to current time.

• The following statement needs more clarity – “Five open-source datasets were used for

map generation and geographical analyses of data [30-34].” What were the nature of these datasets and what sort of geographical analyses were planned?

• The authors have combined a qualitative approach to summarizing the content of articles identified within a scoping review framework. Although this a novel approach, it is undermining the strong methodological framework of pure qualitative research. For e.g., a conceptual or theoretical framework was not predefined, it is not clear whether inductive or deductive analysis was planned before the data extraction began, whether any saturation limit was set for theme identification, whether any pre-defined themes were not at all found in the literature.

• It is not clear whether, the authors have included any grey literature sources such academic student research work, government reports, NGO reports and so on. If not, it is important to justify why and how this could affect the results. In one place they state that they used grey literature but in the limitations they state that they did not search for unpublished literature and so one is confused about which statement could be the correct one.

• How many of the studies were systematic reviews?

• Type of hospital is not reported in a single axis: both level of care and management style have been mixed up. For eg., Tertiary and government hospital are not mutually exclusive. It would be better to separate both these axis and present in Table 1. Similarly, focus of the data mainly talks about study population and the categories are not mutually exclusive, causing confusion in interpretation, for eg., patients can be population too, a more appropriate word would be community.

• How did the authors manage non-English article data extraction and how was it standardised? Were there any Arabic literature?

• The authors began with the premise that officially reported statistics may be different from realities on the ground and set the stage for clarification of this issue. However, neither the results nor the discussion addressed this in any way. The results highlight important areas for knowledge gap and unseen populations but that has nothing to do with discrepancies in reported and actual data. This needs to be addressed in some way in the analysis, perhaps a trend of MMR from the identified articles or an article(s) which highlights this issue. How do the authors substantiate this statement/conclusion – “Three research solutions emerged to broaden the research perspective and reconcile official and local Moroccan MIH realities: increase geographic breadth, address missing topics and populations, and embrace interdisciplinary methods.”

Minor comments

• This line in abstract could be written clearly “Maternal health predominated, often >50% more than infant health.”

• The last date of literature search was Dec 2022 and one year has elapsed since. Therefore, it might be necessary to run the search for 2023 and see if any new articles have come up in the gap areas.

Reviewer #2: The scoping review has been drafted systematically related to MIH in Morocco populations. Search strategies and PRISMA diagram has been defined clearly. However the data extracted only certain parameters like genders, study population (infants), researchers, their policies, success stories, etc.

1. There was a mention about disconnect between official statistics and some local Moroccan health realities. How was the data extraction and analysis handled from the studies during these period.

2. Had the analysis been conducted with similar type of studies, proper statistical analysis could have been applied rigorously. Why was it not done?

3. The title says this paper is a scoping review, then how could some data of network meta-analysis been performed.

4. There is no clear data / tabulation on performance indicators of the policies adopted by the Moroccan health system like MMR, etc.

5. The themes of the research handled only women and infants. Geriatrics or men also form a part of their population. Data on these population may exhibit varied results on the outcomes.

5. Has the protocol of scoping review been published anywhere ?

6. PLOS authors have the option to publish the peer review history of their article (what does this mean?). If published, this will include your full peer review and any attached files.

**Do you want your identity to be public for this peer review?** For information about this choice, including consent withdrawal, please see our Privacy Policy.

Reviewer #1: No

Reviewer #2: **Yes: **Dr Aravinda Kumar B

---

## [Decision Letter · Decision Letter 1]

24 Jun 2024

Mapping Maternal and Infant Health in Morocco: A Global Scoping Review of Themes, Gaps, and the "Unseen" in the Published Health Research Literature, 2000-2022

PGPH-D-23-02479R1

Dear Dr. Amster,

We are pleased to inform you that your manuscript 'Mapping Maternal and Infant Health in Morocco: A Global Scoping Review of Themes, Gaps, and the "Unseen" in the Published Health Research Literature, 2000-2022' has been provisionally accepted for publication in PLOS Global Public Health.

Best regards,

Julia Robinson

Executive Editor

Reviewer Comments (if any, and for reference):

Reviewer's Responses to Questions

**Comments to the Author**

1. If the authors have adequately addressed your comments raised in a previous round of review and you feel that this manuscript is now acceptable for publication, you may indicate that here to bypass the “Comments to the Author” section, enter your conflict of interest statement in the “Confidential to Editor” section, and submit your "Accept" recommendation.

Reviewer #2: All comments have been addressed

Reviewer #3: All comments have been addressed

2. Does this manuscript meet PLOS Global Public Health’s publication criteria? Is the manuscript technically sound, and do the data support the conclusions? The manuscript must describe methodologically and ethically rigorous research with conclusions that are appropriately drawn based on the data presented.

Reviewer #2: Yes

Reviewer #3: Yes

3. Has the statistical analysis been performed appropriately and rigorously?

Reviewer #2: Yes

Reviewer #3: Yes

4. Have the authors made all data underlying the findings in their manuscript fully available (please refer to the Data Availability Statement at the start of the manuscript PDF file)?

Reviewer #2: Yes

Reviewer #3: Yes

5. Is the manuscript presented in an intelligible fashion and written in standard English?

Reviewer #2: Yes

Reviewer #3: Yes

6. Review Comments to the Author

Reviewer #2: The authors have meticulously addressed all the queries raised and it is convincing.

Reviewer #3: No further comments

7. PLOS authors have the option to publish the peer review history of their article (what does this mean?). If published, this will include your full peer review and any attached files.

**Do you want your identity to be public for this peer review?** For information about this choice, including consent withdrawal, please see our Privacy Policy.

Reviewer #2: **Yes: **Aravinda Kumar

Reviewer #3: **Yes: **Rizwan Abdulkader
